# Optimal Policymaking under Yardstick Vote: An Experimental Study

Albert Argilaga [1] and Jijian Fan [2,3,*]

1 College of Civil Engineering and Architecture, Zhejiang University, Hangzhou 310058, China; argilaga@zju.edu.cn
2 School of Economics, Zhejiang University, Hangzhou 310058, China
3 Institute for Fiscal Big-Data and Policy, Zhejiang University, Hangzhou 310058, China
* Correspondence: jijianfan@zju.edu.cn

**Abstract:** We develop a numerical model that simulates the evolution of a virtual population with an incentive and ability-based wage, capital yield from savings, social welfare system, and total income subject to taxation and political turnovers. Meta-heuristics, particle swarm optimization (PSO) in particular, is used to find optimal taxation given the constraints of a plurality democracy with yardstick vote. Results show that the policymaker tends to a taxation system that is highly punitive for a minority in order to win the election by benefiting others. Such decision-making leads to a cyclic taxation policy with high taxation targeting sequential portions of the population.

**Keywords:** voting; yardstick competition; decision-making; optimal tax; simple majority; meta-heuristics; particle swarm optimization

## 1. Introduction

Fiscal policies are one of the main tools for governments to intervene in their economies, and lasting democracy with rationalized taxation is necessary for economic growth [1]. Optimal tax theory has received plenty of attention by both economists and policymakers [2]. Early works can be traced back to Ramsey [3], with a much higher interest in the subject during the last 40 years thanks to the works of Mirrlees [4], which became the dominant approach for theorists. Mankiw et al. [2] present lessons from comparing optimal tax theory to a few decades of OECD tax policy [5]. For example:

- Lesson 2: "The optimal marginal tax schedule could decline at high incomes."
- Lesson 3: "A flat tax, with a universal lump-sum transfer, could be close to optimal."
- Lesson 5: "Optimal taxes should depend on personal characteristics and income."
- Lesson 7: "Capital income ought to be untaxed, at least in expectation."

The selected lessons can identify possible implications of democracy on the optimal tax theory. For example, the Lame-Duck effect [6], the probabilistic voting mode [7], as well as the changes in fiscal rules and public spending before the election [8] are among the studies on democracy and economy interplay.

While a supermajority determination of taxation policy may usually ensure efficiency [9–12] and tax mimic across regions [13,14], the widely observed political business cycle may introduce unwanted fiscal effects on efficiency and equality [15–17]. Yardstick competition is one of the mechanisms first introduced by Shleifer [18] in describing firm competitions, followed by Besley and Case [19], who developed this idea into a political competition theory to explain how voters make decisions based upon incumbent's and neighboring jurisdiction's tax policies. They present a political economy tax-setting model where the voters make comparisons between jurisdictions resulting in a yardstick vote. In a cross-mandate case, the incumbent's current tax policy will be viewed as a "yardstick" for the myopic voters and determines their reference point for the next election. The yardstick

vote may also account for the social welfare services provided by the government and evaluate if the tax revenue has been efficiently spent. A vast amount of literature has found that voters punish incumbents for higher tax rates, depending on government traits and potential policy competitors [20,21].

The yardstick vote introduces strategic policy-making for any neutral government that only cares about the election results. If the incumbent government wants to be re-elected, it may not be enough for them to maintain the current regime that derives the same well-being of the voters. Instead, they need to consider the competing taxation regimes with which any potential political competitors may show up and strategically interact. Therefore, the yardstick competition implies that any government that wants to stay incumbent must keep beating itself rather than keeping the status quo, which leads to sequential policy-making. However, how the yardstick voting affects political turnovers and the corresponding optimal taxation remains unknown.

The motivation of the present paper is to provide a simple yet representative model able to reproduce the fiscal and economic evolution of a society under the constraint of a simple majority democracy and yardstick vote to demonstrate the relation between wealth distribution, policymaking, and voting intention.

We develop a simple model and numerically simulate the taxation regime evolution to answer this question. A virtual population with an incentive and ability-based wage, capital yield from savings, social welfare system, and total income subject to taxation and political turnovers are considered under the parameters calibrating economic growth in the past several decades and the distribution of household wealth in the U.S. (Figure 1). In each mandate, the government determines its optimal income tax regime characterized by a truncated linear function of the marginal tax rate. It then distributes the revenue back to the population equally in public goods. Their relative social ranking determines voters' preferences and political attitudes, and the government's problem is to maximize the voting to be re-elected.

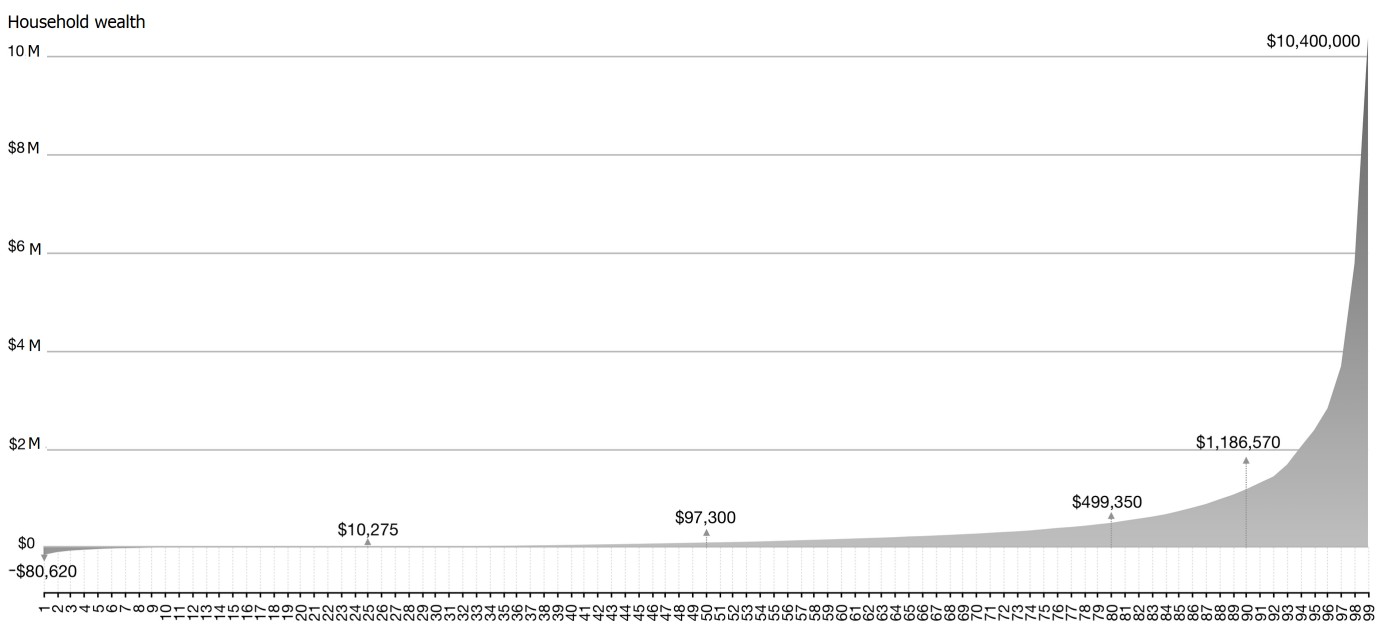

**Figure 1.** Distribution of household wealth in US 2016 [22].

Meta-heuristics method, particle swarm optimization (PSO) in particular, is used to find optimal taxation given the constraints of a simple majority democracy with a yardstick vote. Results show that the policymaker tends to a taxation regime that is highly punitive for a moving minority in order to win the election by benefiting others. Such decision-making leads to a cyclic taxation policy with punitive taxation targeting sequential portions of the population.

The paper is organized as follows. Section 2 details the numerical model, its assumptions, and the numerical values resulting from calibration. Section 3 presents experimental results from the proposed model. Section 4 introduces the meta-heuristics to optimize the problem defined by the model and a yardstick reference. Section 5 performs an experimental study with successive mandates. Section 6 discusses the results, and Section 7 concludes.

## 2. Model

An in-house numerical model with a fixed population is used to simulate the economic evolution of a virtual society, assuming that the individuals of this society can obtain a capital yield from their savings and wage income and are liable to taxation according to their total income. This assumption is chosen as a compromise between Lesson 7 in the literature review [23] and the OECD countries not yet applying a zero capital tax [24]. This taxation mechanism is consistent with most countries not applying tags or discriminating taxation other than by income. For the sake of simplicity, migrations are not allowed in the virtual society and the total population remains constant. Wage income depends on ability and incentive, and the ability is taken constantly along the simulation with a given initial distribution. While macroeconomic aspects are not the final purpose of the model, our main objective is to model wealth distribution with the representative variables of capital yield, tax rates and tranches. The initial capital ($k_0$) of each individual $i$ in the total population $n$ is assumed to follow the additive uniform exponential distribution (AUED) [25]:

$$k_0 = k1 \cdot (k2 \cdot Rndu(0,1) + Rnde(k3)), \tag{1}$$

$Rndu(a,b)$ being a uniform probability distribution between $a$ and $b$, $Rnde(d)$ an exponential probability distribution with mean $d$, $k1$, $k2$ coefficients and $k3$ the mean of the exponential probability distribution defined in Table 1. Similarly, the initial income ($y_0$) of each individual is assumed to follow the AUED distribution:

$$y_0 = y1 \cdot (y2 \cdot Rndu(0,1) + Rnde(y3)), \tag{2}$$

with $y1$, $y2$ being coefficients and $y3$ the mean of the exponential probability distribution defined in Table 1. Mixed distributions are useful to describe variables with components of different nature; in the present case, total income is composed of ability-based income and capital yield. An example of application of the AUED distribution can be found in the modeling of manpower total length of service [26].

Capital ($k$) and income ($y$) coefficients are calibrated to approximately match the distribution of household income in the U.S. in 2019, published by the U.S. Census Bureau (Table 1):

**Table 1.** Coefficients of Equations (1) and (2) calibrated with data of household income in the U.S. in 2019 published by the U.S. Census Bureau [22].

| Coefficient | 1 | 2 | 3 |
|---|---|---|---|
| Capital (k) | $1 \times 10^6$ | 0.05 | 0.95 |
| Income (y) | $2 \times 10^5$ | 0.70 | 0.30 |

The model assumes that individuals are subject to income taxation by the government: the tax rate is a non-decreasing function of their income $y$, and the total amount of tax is a non-increasing function of $y$. The top marginal tax rate is set to zero after a maximum income $y_{t,max}$ consistent with Lesson 2 in the literature review [4]. The lowest income is set to zero to match redistribution policies seen in most OECD countries, this also makes the lowest incomes have a zero marginal tax rate, which is suboptimal according to the theory. Tax rate $t$ is defined in (Equation (3)):

$$t \xrightarrow{f} f(z) = \begin{cases} t_{min} & z < y_{t,min} \\ (z - y_{t,min}) \cdot \frac{(t_{max}-t_{min})}{(y_{t,max}-y_{t,min})} & y_{t,min} \leq z < y_{t,max} \\ t_{max} & y_{t,max} \leq z \end{cases} \tag{3}$$

where $f(z)$ is the taxation function of a variable $z$, $t_{min}$ and $t_{max}$ are the minimum and maximum tax rates, with $t_{min} = 0$, $y_{t,min}$ and $y_{t,max}$ being the income values defining the limits of the progressive taxation tranche. It is assumed that the government adjusts $y_{t,min}$ and $y_{t,max}$ values every year in order to match a given percentile regardless of income level. Reference minimum and maximum percentiles have been taken as 10% and 70%, so that the values $y_{t,min}$ and $y_{t,max}$ closely match U.S. taxation in 2019. The income $y$ in each year is the result of applying an incentive factor function of the tax rates (Equation (4)) to the ability based initial income distribution (Equation (2)):

$$\eta_{inc} = 1 - \xi \frac{\partial t}{\partial y}, \tag{4}$$

where the tax disincentive $\xi$ is a coefficient with a real positive value, $\xi = 0$ meaning no tax rate disincentive. The resulting incentive factor is defined in each tax tranche in (Equation (5)):

$$\eta_{inc} \xrightarrow{h} h(z) = \begin{cases} 1 & z < y_{t,min} \\ 1 - \xi \frac{(t_{max}-t_{min})}{(y_{t,max}-y_{t,min})} & y_{t,min} \leq z < y_{t,max} \\ 1 & y_{t,max} \leq z \end{cases} \tag{5}$$

where $h(z)$ is the incentive function of a variable $z$. A numerical example in Figure 2 illustrates previous equations (Equations (3) and (5)).

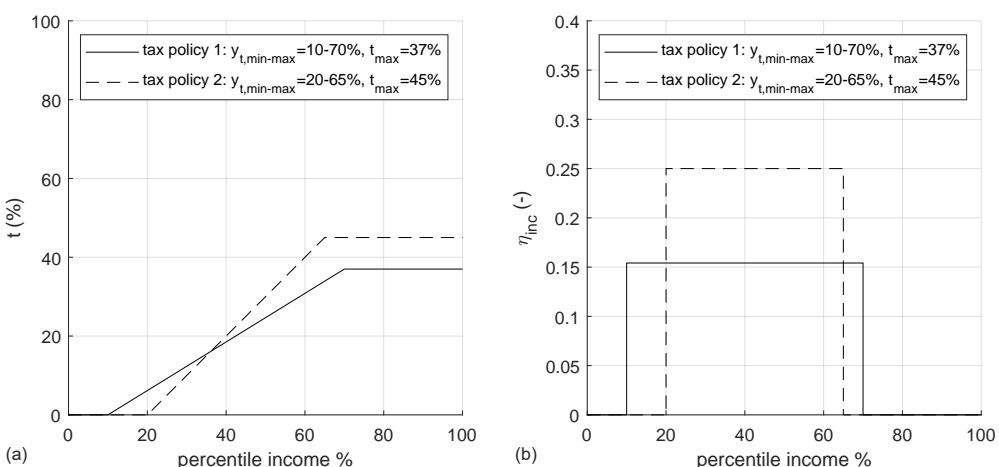

(a)　　　　　　　　　　　　　　　　　　　(b)

**Figure 2.** From left to right: (**a**) Tax rate for two tax policies. (**b**) Efficiency coefficient for two tax policies with $\xi = 0.25$.

The total amount of collected tax $T$ in one exercise (1 year) is defined as (Equation (6)):

$$T = \sum_n t(y_i) y_i \tag{6}$$

and is spent in social welfare in the next exercise, the social welfare expenses are considered to be equally shared among all individuals. An efficiency coefficient $\eta_{wf}$ with a real positive value, $\eta_{wf} = 1$ meaning perfect efficiency, is applied to the social welfare benefits to account for the eventual misuse of resources. An individual expenditure function is used with a constant value of expenses for all the population subject to capital $k$ availability, it is considered that social welfare benefits alleviate the need for individual expenditures. If an

individual has a capital less than the annual expenses minus the social welfare assignation, the expenses are not applied in the present year (Equation (7)):

$$c \xrightarrow{g} g(z) = \begin{cases} 0 & z < c_{th} - \eta_{wf}\frac{T}{n} \\ c_{th} - \eta_{wf}\frac{T}{n} & c_{th} - \eta_{wf}\frac{T}{n} \leq z \end{cases} \tag{7}$$

where $g(z)$ is the expenses function of a variable $z$, $c_{th}$ are the individual expenses with a value $c_{th} = 1.65 \times 10^5$ USD for all individuals, an improvement of the model could consist in the introduction of expenses function of income, for sake of clarity this has not been done in the present research.

Finally, individuals can obtain an extra income from capital yield $r$, a value of $r = 0.0317$ is adopted which is the average of U.S. economy in the period 1947–2020. The model for one individual in one fiscal exercise becomes:

$$k = k_0 + (1 - t(y_i))y_i - c(k_0, \eta_{wf}\frac{T}{n}), \tag{8}$$

where $k_0$ is either the individual's initial capital or the capital resulting from the previous exercise, $y_i$ takes the value of $y_0$ in the initial condition, $k$ is the capital after adding the previous net income according to Equation (3) and subtracting the expenses $c(k_0, \eta_{wf}\frac{T}{n})$ according to Equation (7). The income $y$ is then computed by applying the incentive factor and adding capital yield to the wage based income:

$$y = \eta_{inc}y_0 + k_0 \cdot r, \tag{9}$$

and finally both capital $k_0$ and income $y_i$ are updated for the next exercise:

$$k_0 = k, \tag{10}$$

$$y_i = y, \tag{11}$$

note that capital ($k_0$) is accumulated by individuals during the history of the model, while wage based income ($y_0$) is constant throughout the history as ability is assumed to have a fixed distribution. Equations (8)–(11) are run in nested loops for all the individuals and the successive exercises.

## 3. Validation

### 3.1. Capital and Income Distributions

Histogram representations of Equations (1) and (2) for a population $n = 10,000$ individuals are shown in Figure 3. Capital histogram shows a dominantly exponential distribution with an overall increase of capital due to economic growth during a 73 year period. The income histogram presents a distribution with more influence from the uniform distribution (see the value $y2$ in Equation (2)).

The figure shows that leaving aside economic growth, initial capital and income distributions are qualitatively close to the final distributions calibrated with the distribution of household income in the U.S. in 2019 [22].

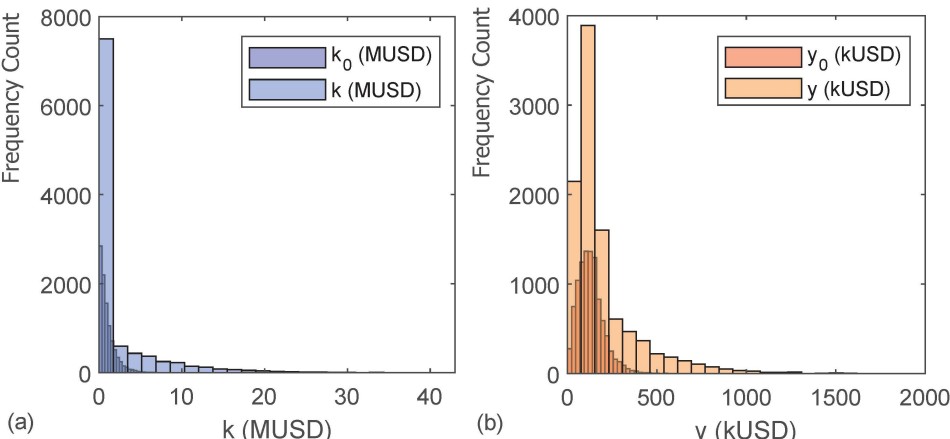

**Figure 3.** Distributions of initial and final capital and income in a $n = 10,000$ population after 73 years with Keynesian tax policy. (**a**) Capital. (**b**) Income.

### 3.2. Taxation Level and Wealth Distribution

In the following, an economy consisting of 10,000 individuals is simulated throughout a 73-year period. The span has been chosen to represent the Fordist industrial system and Keynesian economy from 1947 to 2020. The interest rate $r$ has been taken as the average of this period for the U.S. economy ($r = 3.17\%$). The minimum tax rate $t_{min}$ is zero in all cases. The percentiles to determine $y_{t,min}-y_{t,max}$ are 10–70% in all cases.

Results show that, with no income taxation, individuals below percentile 55% cannot accumulate capital, and their only income comes from wages. In contrast, individuals above percentile 55% have an increasingly big portion of their income coming from capital yield. The portion of population with low income slope with respect to percentile extends up to 50%. Above this value, there is a fast increment of wealth about percentile increment for both income $\Delta y / \Delta p$ and capital accumulation $\Delta k / \Delta p$ (Figure 4). Plots from left to right present: (a) capital and income distributions at the end of the 73 year period and (b) derivative of the capital and income with respect to the income percentile. The semi-logarithmic plot shows that the disparity accelerates with wealth levels approaching percentile 100%.

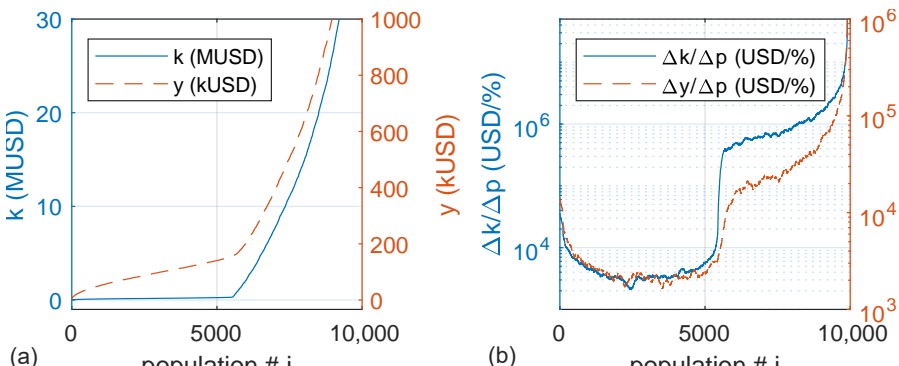

**Figure 4.** Tax rate $t = 0\%$. From left to right: (**a**) Distribution of capital and income in a $1 \times 10^5$ population after 73 years with Keynesian tax policy. (**b**) Differentiation of the wealth with respect to wealth percentile smoothed with a mean window of size 2%.

Figure 5 presents results with varying taxation levels. Plots from left to right present: (a) taxation level, (b) capital and income distributions and (c) derivative of the capital and income with respect to income percentile. Plots from top to bottom rows present different maximum tax rate $t_{max}$ equal to 0%, 37% (U.S. tax system), 45% (typical European country and China) and 56% (Japan). With an income taxation with maximum tax rate $t_{max} = 0.37$ (Figure 5, first row). After percentile 70% disparity growth has a similar shape than the case without taxation.

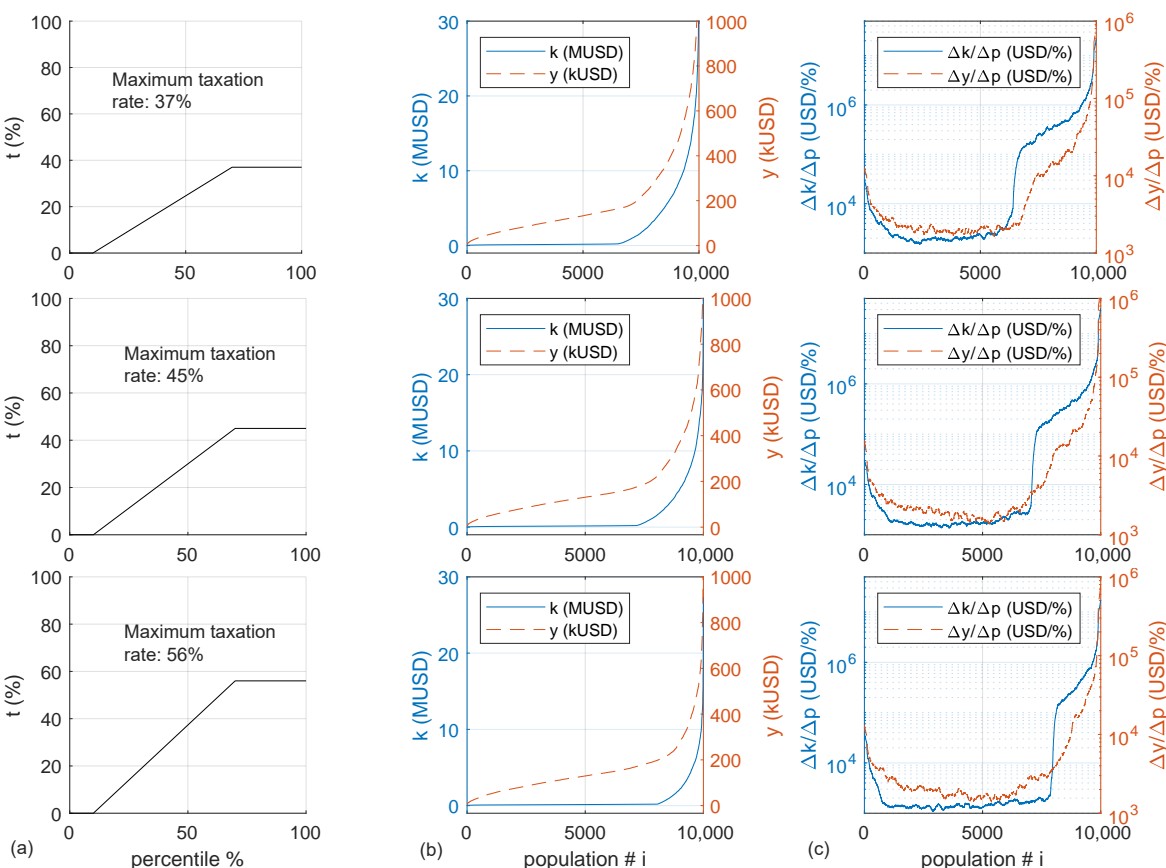

**Figure 5.** From top to bottom: maximum taxation rate of 37%, 45%, 56%. From left to right: (**a**) Tax rate. (**b**) Distribution of capital and income in a $1 \times 10^5$ population after 73 years with Keynesian tax policy. (**c**) Differentiation of the wealth with respect to wealth percentile smoothed with a mean window of size 2%.

With income taxation with a maximum tax rate $t_{max} = 0.45$ (Figure 5, second row), the portion of the population not able to accumulate capital increases up to 75%. Finally, income taxation with a maximum tax rate $t_{max} = 0.56$ (Figure 5, third row) increases the portion of the population without capital accumulation up to 80%, and the economic disparity among the wealthiest is still growing acceleratedly despite the higher tax.

### 3.3. Taxation Range and Wealth Distribution

The previous subsection shows simulations with a progressive taxation range from percentile 10% to 70%. In the following figure, results with a translation of the progressive taxation range are shown keeping a maximum taxation $t_{max} = 37\%$ (Figure 6).

Progressive taxation range translation to percentiles 20–80% shows an increase of individuals able to accumulate capital (Figure 6, first row), this indicates an increase of individuals with higher income. Progressive taxation range translation to percentiles 30–90% shows a further increase of individuals able to accumulate capital (Figure 6) and increases disparity $\Delta y / \Delta p$ and $\Delta k / \Delta p$ now in the range 40–70%.

Results show that an increasing maximum tax reduces the saving capacity of an increasing number of individuals. Translation of the tax tranches to a higher income level (overall reduction of tax) has the contrary effect in addition to introducing a disturbance in the $\Delta y / \Delta p$ values in the middle of the percentile range.

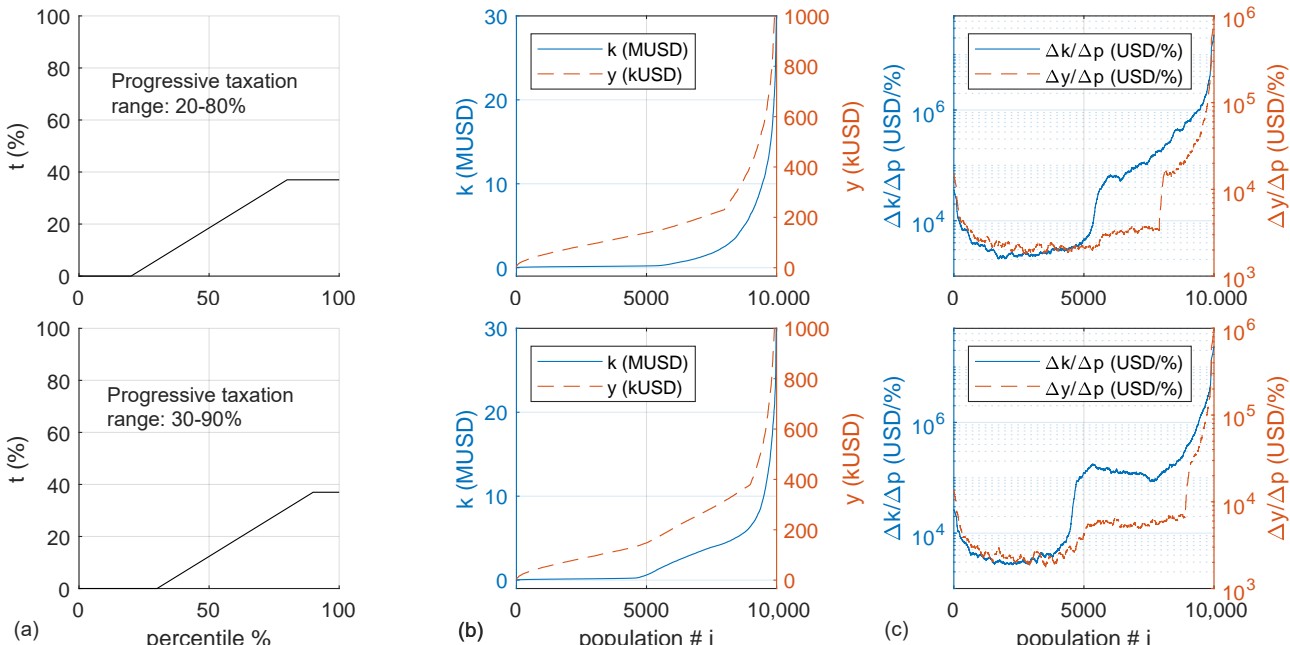

**Figure 6.** From top to bottom: Progressive taxation range: percentiles 20–80 and 30–90. From left to right: (**a**) Differentiation of the tax rate with respect to wealth percentile. (**b**) Distribution of capital and income in a $1 \times 10^5$ population after 73 years with Keynesian tax policy. (**c**) Differentiation of the wealth with respect to wealth percentile smoothed with a mean window of size 2%.

## 4. Meta-Heuristics Based Policymaking: Optimization

The previous section explores the effects of taxation on wealth distribution in the full spectrum of society. Given the constraint of democracy with a plurality vote, the incumbent government's optimization problem is to minimize the population unhappy with the newly proposed economic policy. Therefore, an income-based yardstick vote [19–21], is used as a criterion to measure voters' intentions. Contrary to strategic voting [27,28], in the present model, voting is assumed to be sincere: the vote is for the policy that maximizes its interest, and voters do not have information about other voter's intentions. With this logic, voters evaluate the incumbent's policy at the end of a mandate. The other candidates never come into play because it is assumed that the incumbent always adopts the optimal policymaking regarding re-election. The model does not account for population fluxes; while migrations can have a major impact on demographics and aggregate productivity at long term, in the present work it is considered that short term policymaking perceives potential migrations as lost or gained votes.

Due to the non-linearity, non-derivability, and presence of multiple local minima of the problem, classical and gradient-based optimization methods cannot be used. Particle swarm optimization (PSO), first developed by Kennedy and Eberhart [29] and Eberhart and Kennedy [30], is adopted as a derivative-free meta-heuristics optimizer. The reasons for PSO popularity can be numerous; its simplicity and adaptability to a multitude of real-world problems are among the most appealing features. In the present case, and because of the unknown nature of the problem resulting from a yardstick process, the ability to tune the exploration–exploitation bias of the algorithm has been determining. Using a meta-heuristic with poor exploration capability could lead to local optima trapping. PSO allows us to initially tune the algorithm to ensure a thorough search space exploration, once more information about the global optimum position is obtained exploitation can be increased back to gain performance. In this work, the later improvement inertia-PSO or $\omega$PSO [31] is adopted. This version of the algorithm gives inertia weight to the particles, which improves the exploitation phase during the search.

The optimization variables being $y_{t,min}$, $y_{t,max}$ and $t_{max}$, they are grouped in this order in the vector $x$. Results of the model in the previous section after a 73 year run with

$x = [0.10\ 0.70\ 0.37]$ are used as initial conditions. One simulation is run for a given period with the same values of $x$ as in the 73 year pre-initial condition period, serving as a yardstick reference. The optimization is performed in an incumbent run with the same initial conditions as the yardstick reference and with the same time length. At the end of the incumbent run, each $i$ of the population $n$ is compared against its yardstick reference homologous, and the income level $y$ is used as a criterion to determine vote intention; if the income $y$ is lower compared to the yardstick reference, yardstick vote punishes the incumbent. The length of the incumbent run is taken equal to four years, coinciding with a usual electoral mandate. The optimization problem to be solved reads:

$$min \quad i_{ys}/n, \tag{12}$$

subject to:

$$
\begin{aligned}
y_{t,min} &\in [0.0, 100.0]\% \\
y_{t,max} &\in [0.0, 100.0]\% \\
t_{max} &\in [0.0, 100.0]\%
\end{aligned}
\tag{13}
$$

where $i_{ys}$ is the number of individuals unhappy with the economic policy according to the yardstick vote. Note that the ranges of the optimization variables are all $[0.0\ 100.0]$, while $y_{t,min}$ is never larger than $y_{t,max}$. To avoid search in the infeasible space, a penalization method is adopted [32]. Previous minimization (Equation (12)) becomes:

$$min \quad i_{ys}/n + \lambda(y_{t,min} - y_{t,max}) \tag{14}$$

where $\lambda(y_{t,min} - y_{t,max})$ is the penalty term with $\lambda$ the penalty coefficient defined by the function $q$:

$$
\lambda \xrightarrow{q} q(z) = \begin{cases} 100 & y_{t,max} - y_{t,min} < 1 \\ 0 & y_{t,max} - y_{t,min} \geq 1 \end{cases}
\tag{15}
$$

where a value of 1% is selected as the minimum distance between $y_{t,min}$ and $y_{t,max}$ to avoid infinite tax gradients. The parameters used in the PSO algorithm are intended to maximize exploration, thus a relatively high swam size is used, inertia range is clipped at 1.0 to avoid premature convergence and the self adjustment weight (cognitive component) is set to twice the population adjustment weight (social component). Table 2 presents the set of PSO parameters prior to swarm size determination:

**Table 2.** Particle swarm optimization (PSO) parameters.

| PSO Parameter | Value |
|---|---|
| Function Tolerance | $1.0 \times 10^{-6}$ |
| Inertia Range | $[0.1000, 1.0000]$ |
| Min. Neighbors Fraction | 0.25 |
| Objective Limit | 0.0 |
| Maximum Iterations | 50 |
| Maximum Stall Iterations | 10 |
| Self Adjustment Weight | 2.0000 |
| Population Adjustment Weight | 0.9900 |

A parametric study on the swarm size is performed, and the swarm sizes are chosen to be multiples of 8 to maximize the performance of the 8 thread parallelization. Only one optimization result is shown for each swarm size (Table 3). The two simulations with swarm sizes below 32 do not present a good convergence on $f(x)$. Starting at swarm size 32 and up, the best $f(x)$ are convergent and the value of the optimized $x$ tends to a global optimum close to $x = [0.1287\ 0.6324\ 0.3636]$, which is the value with the lowest $f(x)$ (simulation 4).

**Table 3.** Particle swarm optimization summary. Run in a computer with Intel Core i7-10700 processor @2.90 GHz and 16.0 GB RAM @2933 MHz, all simulations using 8 thread parallelization. * Value falling in a limit of the optimization ranges (13).

| Sim. | Swarm Size | $t_{max}$ | Best $f(x)$ | Mean $f(x)$ | Optimal x | CPU Time |
|---|---|---|---|---|---|---|
| 1 | 8 | 41% | 0.1088 | 0.2036 | [0.0357 0.7719 0.4103] | 154″ |
| 2 | 16 | 41% | 0.1355 | 0.1435 | [0.0916 0.1686 0.4121] | 250″ |
| 3 | 32 | 37% | 0.1049 | 0.2338 | [0.1579 0.6036 0.3747] | 429″ |
| 4 | 64 | 36% | 0.1013 | 0.2303 | [0.1287 0.6324 0.3636] | 1172″ |
| 5 | 128 | 36% | 0.1038 | 0.2424 | [0.200 * 0.5570 0.3667] | 1475″ |
| 6 | 256 | 37% | 0.1025 | 0.2206 | [0.1031 0.6712 0.3758] | 3668″ |
| 7 | 512 | 36% | 0.1021 | 0.2330 | [0.200 * 0.5475 0.3566] | 4196″ |

Given previous experimental outcomes and the available computational resources, a swarm size 120 is selected. A series of simulations are used to test the convergence to a unique solution and validate the PSO parameters. A total of 30 optimizations give a mean best target function $f(x) = 0.0874$ with standard deviation $f(x)\sigma_{n-1} = 0.0334$. The small standard deviation for the mean provides reasonable confidence about the results using a swarm size of 120. A parametric study on welfare efficiency and work incentive with values in the ranges $\eta_{wf} = [1\ 0.8]$ and $\xi = [0\ 0.08]$ is performed for one mandate (4 years) and a selection of the results is presented in Table 4.

**Table 4.** PSO summary for one mandate (4 years).

| $\eta_{wf}$ | $\xi$ | Optimal x |
|---|---|---|
| 1 | 0 | [0.0938 0.9961 0.9662] |
| 1 | 0.002 | [0.1160 0.9941 0.9247] |
| 1 | 0.004 | [0.1357 0.9936 0.9347] |
| 1 | 0.006 | [0.2676 0.7023 0.3366] |
| 1 | 0.008 | [0.2191 0.7007 0.3443] |
| 1 | 0.010 | [0.2241 0.7020 0.3443] |
| 0.8 | 0 | [0.1833 0.7336 0.3454] |
| 0.8 | 0.002 | [0.1266 0.6997 0.3615] |
| 0.8 | 0.004 | [0.1590 0.6988 0.3498] |
| 0.8 | 0.006 | [0.1840 1.6944 0.3387] |
| 0.8 | 0.008 | [0.2083 0.6948 0.3295] |
| 0.8 | 0.010 | [0.2395 0.6945 0.3141] |

Welfare efficiency $\eta_{wf} = 1$ gives a non-zero top-marginal rate up to tax disincentive $\xi = 0.004$, suggesting that in real economy, at least one of the conditions $\eta_{wf} < 1$ or $\xi > 0.004$ is found. Results with $\eta_{wf} < 1$ show a progressive decrease of optimal taxation overall income $y$ range for an increasing value of $\xi$ starting at $\xi = 0$. The same selected results of the parametric study are presented in Table 5 for two consecutive mandates (8 years). A full parametric study including 30 results for each case of one and two mandates can be found in the repository: https://doi.org/10.6084/m9.figshare.16442916.v3 26 August 2021—First online date, Posted date, Accessed date).

For a welfare efficiency $\eta_{wf} = 1$ taxation decrease is observed for all values of $\xi$, with $y_{t,min}$ increasing up to approx. 50% in all the range. For suboptimal welfare efficiencies ($\eta_{wf} < 1$) taxation reductions are larger and present non-zero top-marginal rate ($\eta_{wf} = 0.9$) and tend to zero taxation for $\eta_{wf} = 0.8$ (Table 5).

Two optimization cases are shown for the case $\eta_{wf} = 1$ and $\xi = 0.006$ for one mandate (Figure 7, top row) and two mandates (Figure 7, bottom row), this case appears to be close to real economies. Convergence presents the decrease of the best evaluation of the target function (Equation (12)), fitness is the evolution of the average value of target function evaluations in the swarm, and the scatter plots present the history of particle positions in two planes of the optimization space $x$.

**Table 5.** PSO summary for two mandates (8 years). * Value falling in a limit of the optimization ranges (13).

| $\eta_{wf}$ | $\xi$ | Optimal x |
|---|---|---|
| 1 | 0 | [0.5097 0.7780 0.2810] |
| 1 | 0.002 | [0.4879 0.8216 0.2837] |
| 1 | 0.004 | [0.5433 0.8405 0.2718] |
| 1 | 0.006 | [0.5208 0.8690 0.2702] |
| 1 | 0.008 | [0.4799 0.9197 0.3017] |
| 1 | 0.010 | [0.5267 0.7779 0.2442] |
| 0.8 | 0 | [1.000 * 1.000 * 0.0576] |
| 0.8 | 0.002 | [1.000 * 1.000 * 0.3076] |
| 0.8 | 0.004 | [0.6876 1.000 * 0.0232] |
| 0.8 | 0.006 | [1.000 * 1.000 * 0.0819] |
| 0.8 | 0.008 | [0.6206 1.000 * 0.0056] |
| 0.8 | 0.010 | [0.8198 0.9442 0.000 *] |

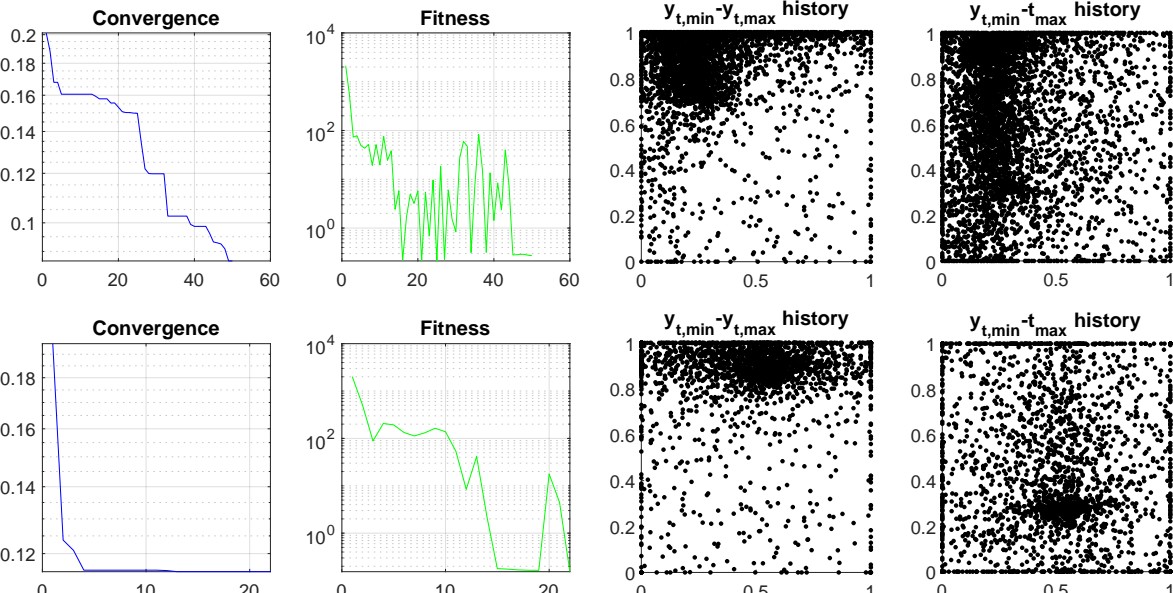

**Figure 7.** From left to right: blue plot is the convergence of the PSO algorithm (minimum value of $f(x)$), green plot is the fitness of the population (mean value of $f(x)$) both plots horizontal axes are the PSO iterations. The two scatter plots represent different planes in the optimization space $x$, $y_{t,min}$-$y_{t,max}$ and $y_{t,min}$-$t_{max}$ respectively. Axes are dimensionless. Swarm Population: 120. **Top line**: one mandate (4 years), **bottom line**: two mandates (8 years). Full results in the repository: https://doi.org/10.6084/m9.figshare.16442916.v3 (26 August 2021—First online date, Posted date, Accessed date).

The same two cases are presented in terms of tax policy and wealth distribution for one mandate (Figure 8, top row) and two mandates (Figure 8, bottom row).

Tax tranches show the lower optimal taxation for longer mandates for the yardstick reference. The consequence of this policy is a slight displacement towards the low rents of the high values of $\Delta k / \Delta p$ around the percentile $p = 40\%$. This is due to an increase in the population that accumulates capital and votes for the incumbent, while losing votes from the lowest end due to a shrinking welfare system.

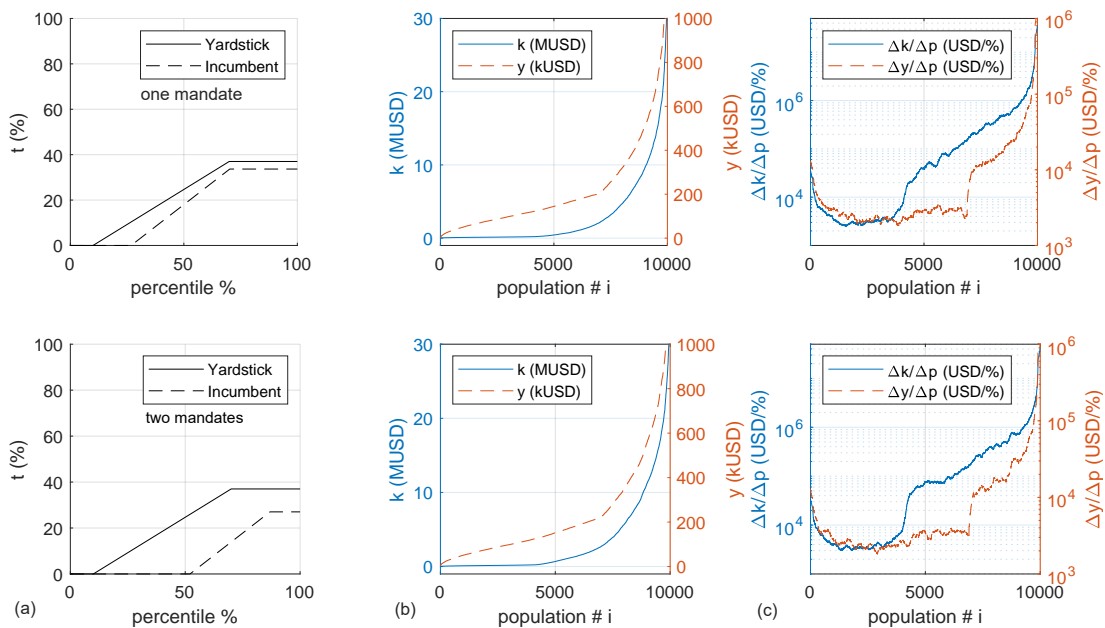

**Figure 8.** Top row: one mandate, bottom row: two mandates. From left to right: (**a**) Tax rate. (**b**) Distribution of capital and income in a $1 \times 10^5$ population after 73 years with Keynesian tax policy. (**c**) Differentiation of the wealth with respect to wealth percentile smoothed with a mean window of size 2%. Full results in the repository: https://doi.org/10.6084/m9.figshare.16442916.v3 (26 August 2021—First online date, Posted date, Accessed date).

## 5. Recursive Mandates

The previous section assumes a unique optimization with the initial conditions established by calibrating the model with real-world data. This section implements a recursive optimization, and the initial conditions are identical. The optimization uses a yardstick reference from the previous recursive optimization at each mandate. The use of a unique reference during the optimization eliminates the possibility of a Condorcet paradox in the voting, which would result in a non-converging optimization. In addition, the order of Equations (7) and (8) is reversed in the algorithm sequence, which means that now all the tax collected in one year is spent in the same year. This eliminates expenditure bias in short mandates seen in Figure 8, thus eliminating mandate length influence on policymaking.

### 5.1. Median Criterion

In this case, the objective is not to minimize discontent, but to maximize the economic improvement of the individual in the median of the population; this is not a realistic objective, but serves to understand the effect of a succession of governments that target a specific economic group in their planning. The optimization problem becomes Equation (16), with the same bounds as in Equation (13).

$$min \quad 1 - \left( \frac{median(\Delta y_i)}{median\left( y_i^{t-1} \right)} \right) + \lambda(y_{t,min} - y_{t,max}), \tag{16}$$

where $\Delta y_i$ is the income increase of individual $i$ with respect to the mandate $t - 1$. The median value of the income in the previous mandate $y_i^{t-1}$ is used to normalize the criterion. A parametric study is performed for a series of welfare efficiencies $\eta_{wf} = [1.0\ 0.95\ 0.9\ 0.8\ 0.7\ 0.5\ 0.4\ 0.25\ 0.1\ 0.0]$ and tax disincentives $\xi = [0.0\ 0.3\ 0.5\ 0.7\ 1.0]$; eight recursive mandates are optimized in each case. Results are presented in Appendix A, Figures A1 and A2.

Cases with $\eta_{wf} = 0.0$ give a result with zero tax for all recursive mandates, and this is not out of expectation because $\eta_{wf} = 0.0$ means that all tax revenues are misused and do not benefit the population at all. In all the other cases, the tax policy evolution presents oscillations with a period between two and five mandates. The oscillations can be pretty dramatic. For example, in the case $\eta_{wf} = 0.9$ and $\xi = 0.3$ from mandate two to three, the maximum tax $t_{max}$ is reduced from 100% to 25% in addition to increasing the length of the progressive tax tranche. Similar scenarios can be observed across the parameter range. These results state that targeted social policymaking combined with yardstick vote does not converge to a stable policy under the different assumptions regarding public spending efficiency and work incentive.

To intuitively understand this phenomenon, we can imagine an incumbent government that, at each mandate, economically sacrifices part of the wealthiest population and rewards the remaining citizens with an income in the percentile 50%. This policy will increase wealth in the middle of the spectrum with a flattened income curve. Once the income distribution is flat between percentiles 50% and 100%, the government needs to target a new group of citizens from lower-income levels to become the new targeted group.

This result can be observed in several cases of the parametric simulations like in $\eta_{wf} = 0.25$ and $\xi = 0.3$, where the incumbent progressively reduces $y_{t,min}$ and $y_{t,max}$ during three mandates until there is a policy restart and the process repeats over. This strategy is unlikely to be applied in practice because knowing that they can become tax victims in the next mandate, people would probably identify the pattern, thus losing trust in the incumbent. We add a new requirement to the yardstick vote to account for such adaptive learning. The new criterion penalizes the incumbent if the 5% of individuals most affected by a new tax policy have their incomes reduced by 5% or more. This criterion is added as a penalization method in the minimization problem of Equation (16). Another $\eta_{wf} - \xi$ parametric study is done in this case with ten recursive mandates, results are presented in Figures A3 and A4.

Even if the new criterion stabilizes some trends, periodic policymaking still happens in most simulations. One case in particular, ($\eta_{wf} = 0.1$ and $\xi = 0.3$), which does not show periodic policymaking in a ten mandate period, is studied for a long run with 50 mandates (Figure 9).

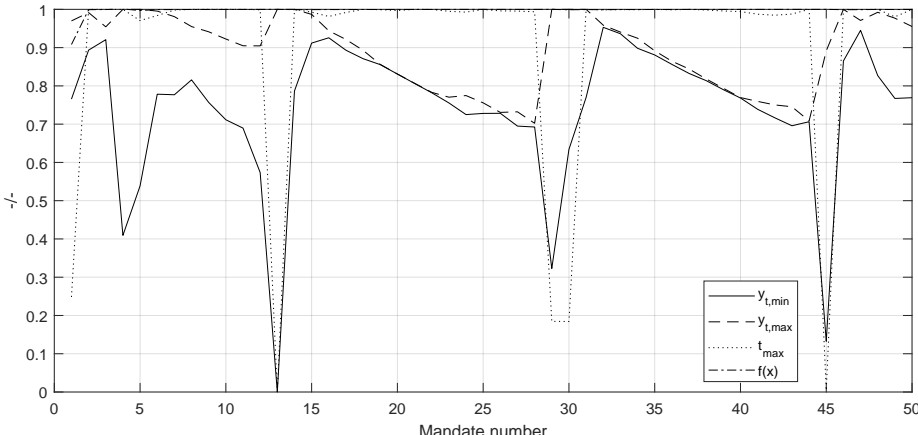

**Figure 9.** Tax policy long run evolution under yardstick vote during ten mandates for welfare spending efficiency $\eta_{wf} = 0.1$ and tax incentive $\xi = 0.3$. Total of 50 four year mandates.

Results show that even if not seen in the short term, the same periodic policymaking as before is observed in the long run with approximately 15 mandates. The incumbent government is still able to obtain a budget surplus by systematically highly taxing a new portion of the population at each new mandate when the demographics start to be insufficient (see mandates 12, 29 and 45 in Figure 9), the incumbent restarts the cycle. These results show that even if the individuals can detect and punish inconsistent government

policymaking, governments can adapt the strategy to maintain tax punishment below the acceptance threshold, thus resulting in a more extended period of cyclic policymaking.

### 5.2. Per Citizen Yardstick Criterion

The parametric study is repeated in this case using the per household yardstick criterion based on income from Equation (12), meaning that the incumbent does not incline to favor any economic group, and its only objective is to win the next election. In the first case, $i_{ys}$ is calculated assuming that citizens will support the incumbent government as far as their income is not decreased for the yardstick (Figure A5). In the second case, the citizens' expectation is more strict for the yardstick and will support the incumbent government only if their incomes are 1% above the yardstick (Figure A6). The first case (non-demanding voters) presents a low value of the objective function $f(x)$, corresponding to the voter's discontent proportion. This voter behavior leads to a trivial constant policy (0% or 100% taxation for the entire population). In the case of demanding voters, however, policymaking becomes a cycle in which sequential parts of the social spectrum are successively punished with high taxation until the cycle restarts. Previous two parametric studies are repeated with a double-length (8 years) mandate (Figures A7 and A8). Longer mandates indicate a tendency to elongate the period of policymaking fluctuations in the case of demanding voters. Nevertheless, the 8-year mandate does not eliminate the fluctuations.

The demanding voter case with a 4-year mandate period (Figure A6) and parameters $\eta_{wf} = 1$ and $\xi = 0$ is used to exemplify the effect of policymaking on the voter spectrum; government policy systematically expands the highly taxed individuals to the lower rents until it is no longer possible to keep voter discontent below 50%, then the cycle restarts (Figure 10).

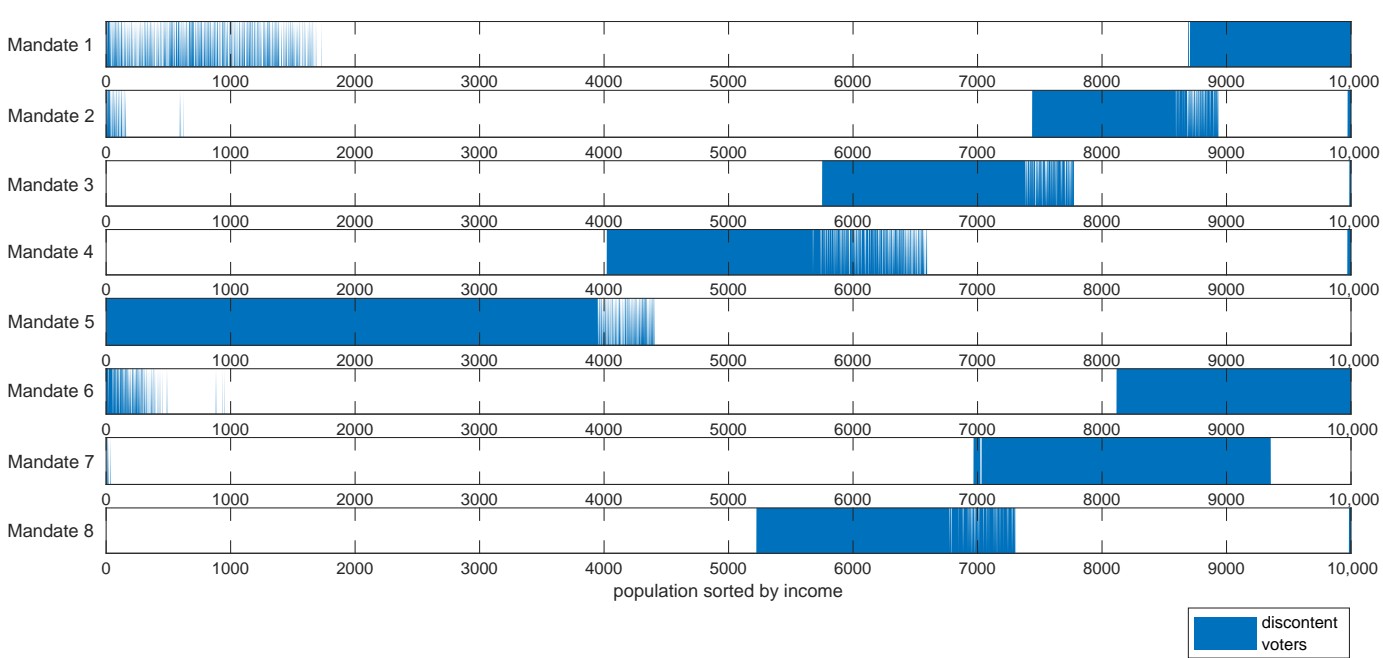

**Figure 10.** Individual discontent in a sequence of eight recursive mandates (from top to bottom). $\eta_{wf} = 1.0$, $\xi = 0.0$. Mandate length: 4 years. Voters income advantage exigency 1%.

## 6. Discussion

The main focus of the work is to study the yardstick vote, wealth distribution, and policymaking. Notably, the model is not based on a general equilibrium concept but on a household-based population, with the economic growth as a calibrated parameter instead of a variable.

The validation using initial and final wealth and income histograms taken from real economies confirms the ability of the model to predict wealth distribution in a period

of 73 years. This setup allows the model to experimentally study the relation between policymaking, voting, and wealth distribution for a short number of mandates.

Results show that a progressive increase of taxation for the income percentile can flatten the income curve and increase the extent of the population not being able to accumulate capital. Regardless of the maximum taxation level in the progressive tranche, once the capital yield can outpace the maximum taxation income $y_{t,max}$, the wealth distribution in this region becomes the same as in a zero tax case. This finding highlights that the flat marginal rate for the higher rents recommended by the optimal tax theory increases the population's economic disparity.

Meta-heuristics optimized tax policy for one single mandate gives a progressive tax tranche with flat marginal rates in both income extremes for any case with welfare efficiency $\eta_{wf} < 1$ and work disincentive $\xi > 0$. This is in agreement with the results of the optimal tax theory for the high rents and shows that in the ideal case with $\eta_{wf} = 1$ and $\xi = 0$ the progressive tax tranche would span most of the population spectrum. Optimization of recursive mandates using a yardstick vote criterion implies that the policymaker must constantly change the tax policy to obtain a majority in the vote. This result holds in a variety of assumptions, i.e., policymaking targeted at the median income or per citizen, income advantage required by voters 0% or 1%, mandate length 4 or 8 years, even in the case of an explicitly formulated rejection of such strategy by the voters.

The obtained tax policy strategy does not fit the recommendations of the optimal tax theory nor the common sense of social equity. It is yet to be determined the effects of such a strategy on economic growth. Combining the present results with a global equilibrium model can help answer this question. Another question is the system's fairness in the long run; after the tax policy has cycled through the income spectrum several times, does it result in an equally tax-punished society? Further research should also study the effect of inter-mandate memory, strategic vote, and other voting rules such as supermajority.

## 7. Conclusions

We develop a numerical model to find optimal taxation given the constraints of a simple majority democracy with a yardstick vote. Results show that the policymaker tends to cyclically impose high taxation and harm a small fraction of the population while using the collected tax to benefit the rest. The presented model has been designed to explain the implications of politics in the context of democracy and optimal taxation policies. The following conclusions are derived from the results:

- Regardless of the maximum taxation level in the progressive tranche, once the capital yield can outpace the maximum taxation income $y_{t,max}$, the wealth distribution in this population region becomes the same as in a no-tax case.
- Meta-heuristics can be an effective tool in policymaking because of their efficiency at finding the optimum of problems with multiple local minima.
- The obtained optimal tax policy for one single mandate is consistent with the recommendations of the optimal tax theory.
- Recursive mandates under yardstick vote lead to cyclic policymaking; the policymaker systematically changes the population's highly taxed portion to assure a simple majority in the next election.

**Author Contributions:** Conceptualization, A.A.; methodology, A.A. and J.F.; software, A.A.; validation, A.A.; formal analysis, A.A. and J.F.; writing—original draft preparation, A.A. and J.F.; writing—review and editing, A.A. and J.F. All authors have read and agreed to the published version of the manuscript.

**Funding:** This research received no external funding.

**Institutional Review Board Statement:** Not applicable.

**Informed Consent Statement:** Not applicable.

**Data Availability Statement:** Data is contained within the article and publicly accessible repository FigShare at https://doi.org/10.6084/m9.figshare.16442916.v3 (26 August 2021—First online date, Posted date, Accessed date).

**Conflicts of Interest:** The authors declare no conflict of interest.

## Appendix A. Parametric Study

Recursive mandate optimization results, parametric study on welfare efficiency $\eta_{wf}$ and tax disincentive $\xi$. With yardstick vote using a median income criterion or an individual voter criterion, voter improvement demand 0% or 1% and mandate length 4 or 8 years.

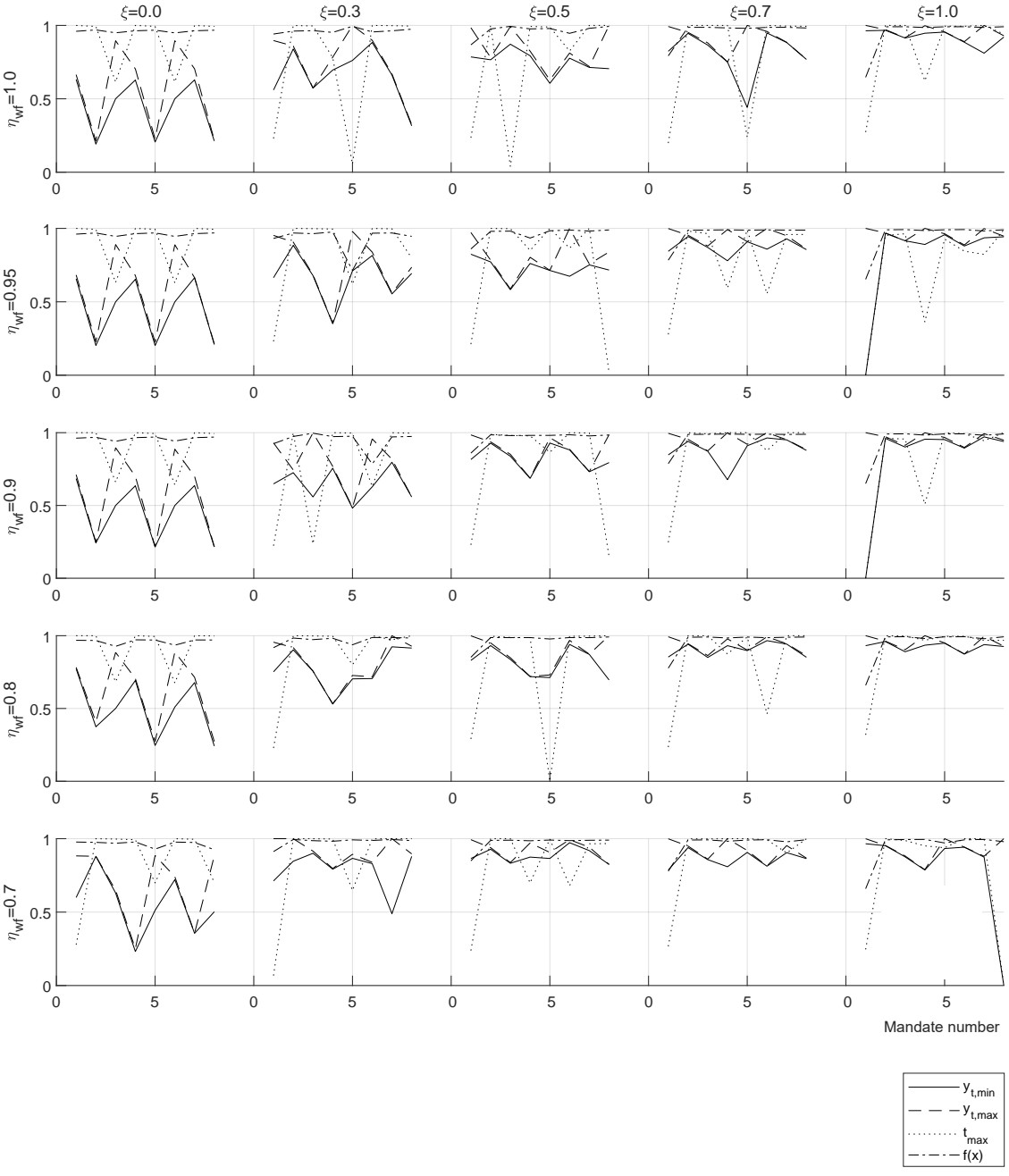

**Figure A1.** Tax policy evolution under yardstick vote during eight mandates for a series of welfare efficiency values $\eta_{wf}$ and tax disincentive $\xi$. Median criterion. Mandate length: 4 years.

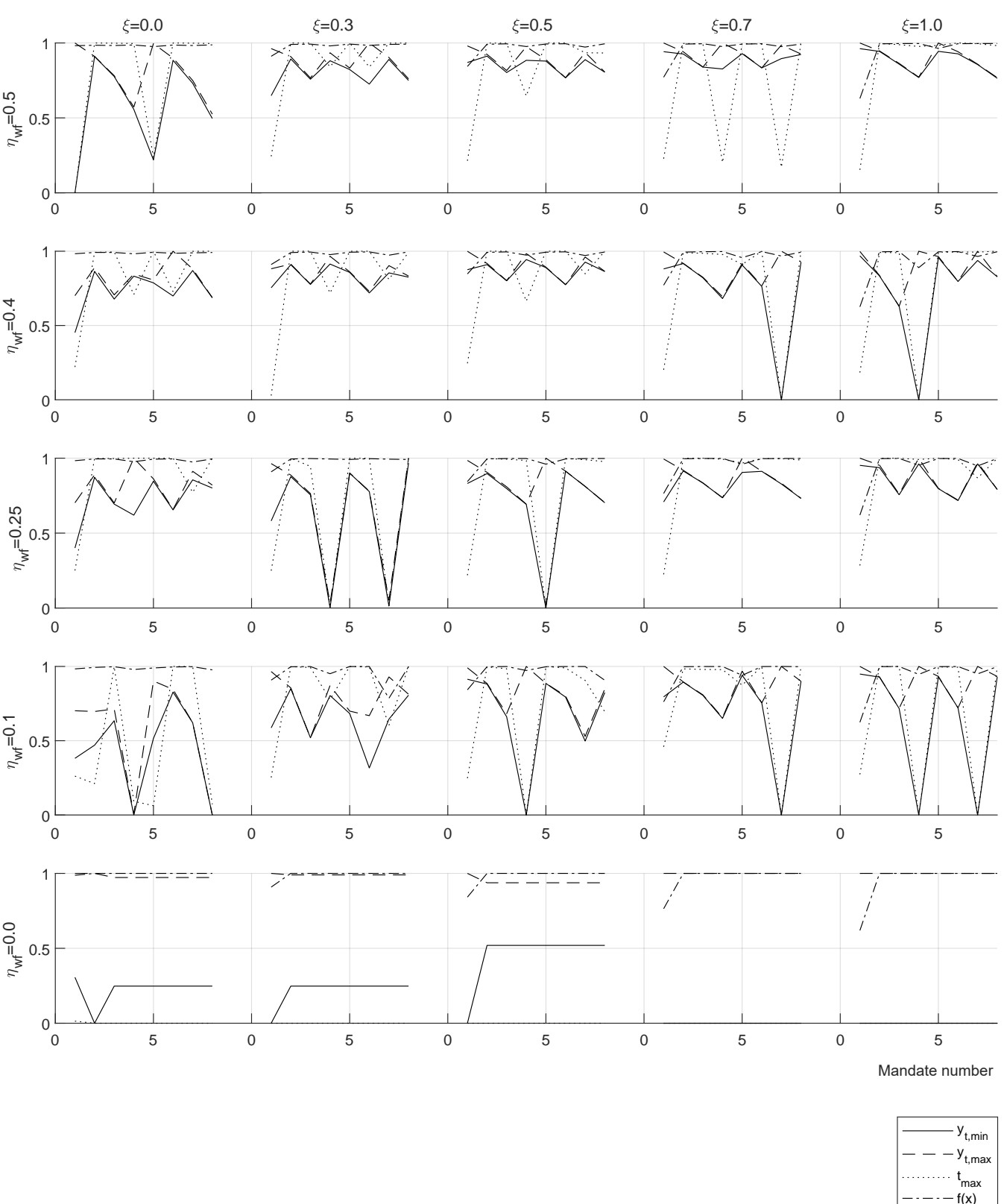

**Figure A2.** Tax policy evolution under yardstick vote during eight mandates for a series of welfare efficiency values $\eta_{wf}$ and tax disincentive $\xi$. Median criterion. Mandate length: 4 years.

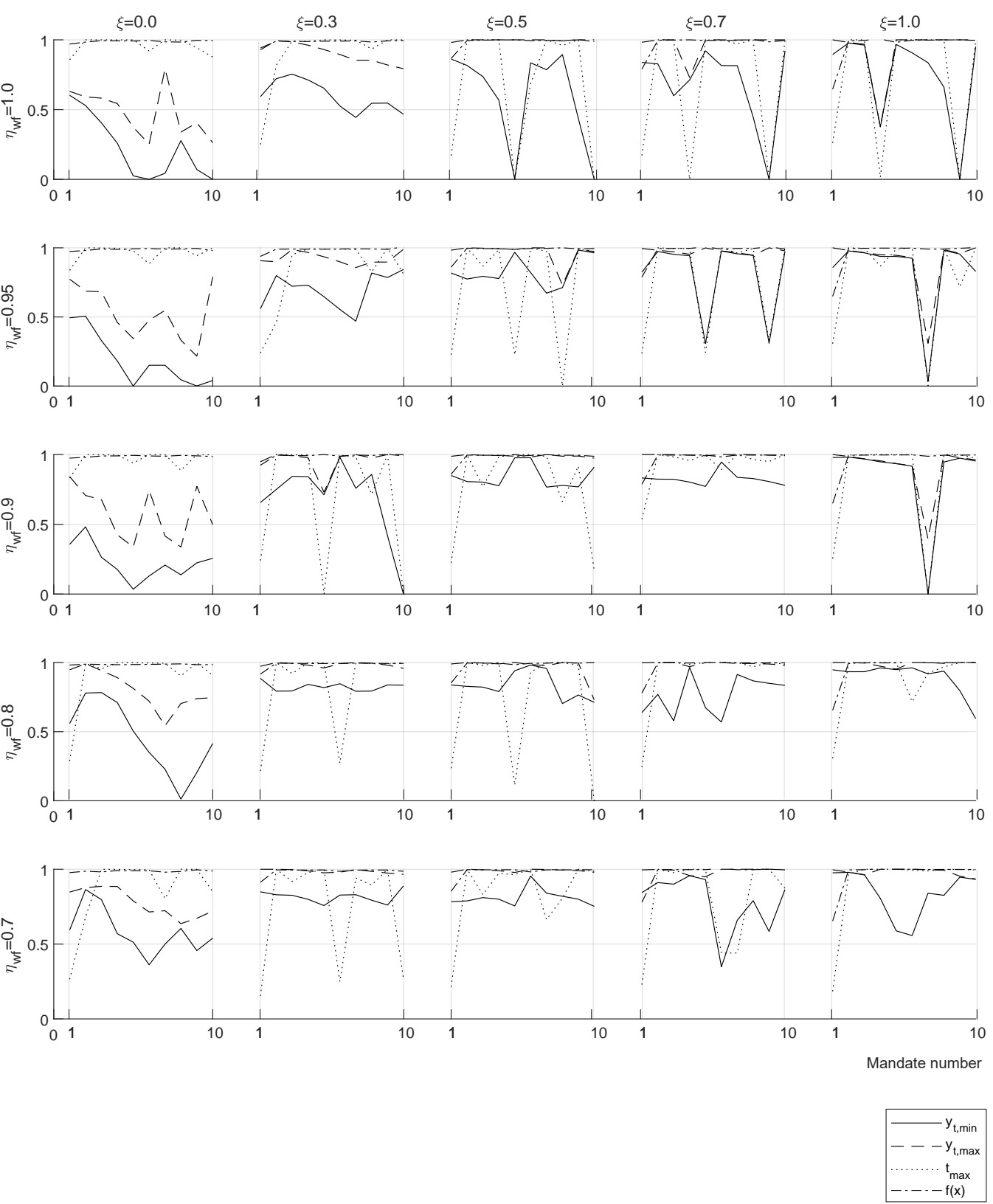

**Figure A3.** Tax policy evolution under yardstick vote during ten mandates for a series of welfare efficiency values $\eta_{wf}$ and tax disincentive $\xi$. Median criterion. Mandate length: 4 years. Vote constrained by stability.

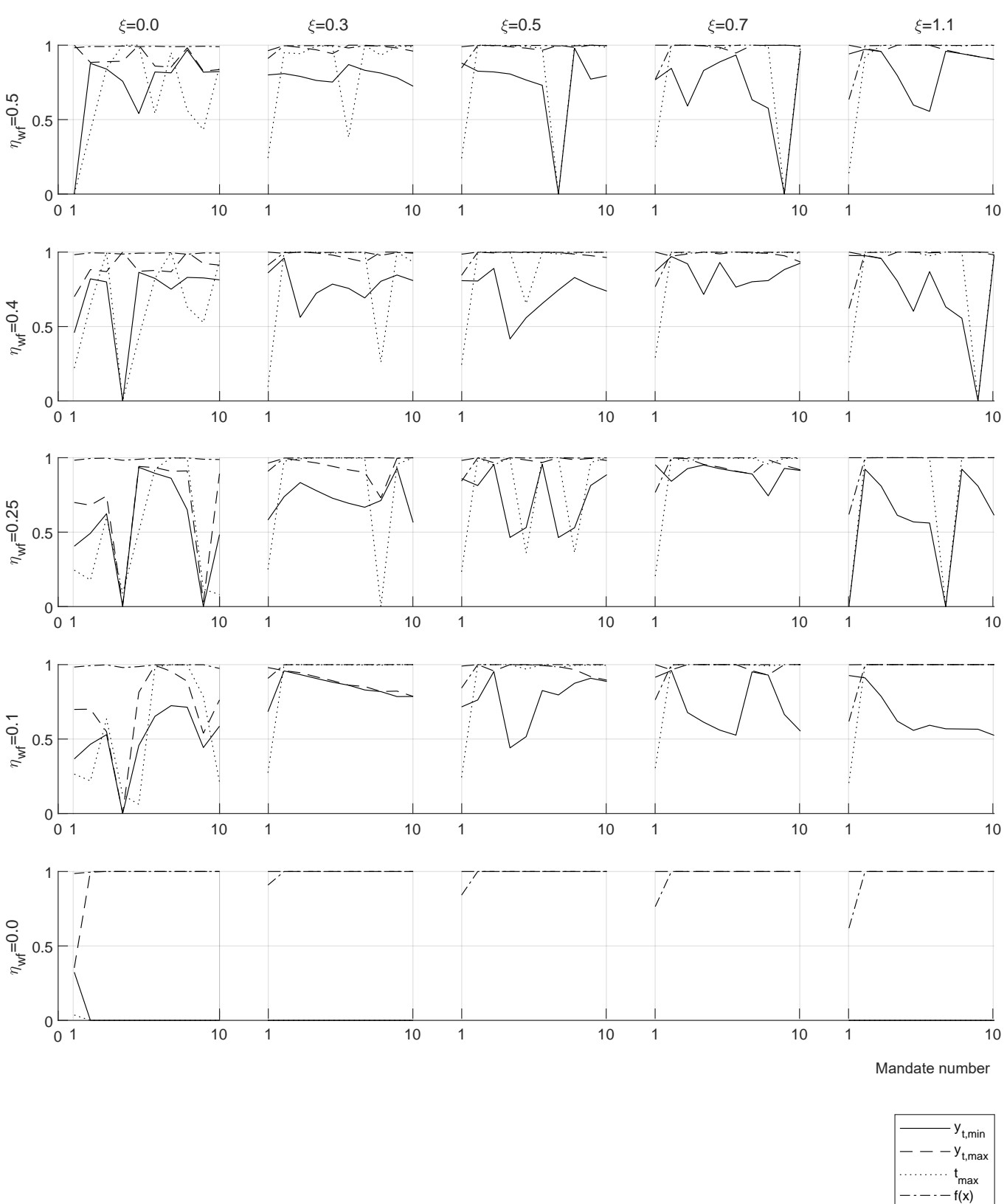

**Figure A4.** Tax policy evolution under yardstick vote during ten mandates for a series of welfare efficiency values $\eta_{wf}$ and tax disincentive $\xi$. Median criterion. Mandate length: 4 years. Vote constrained by stability.

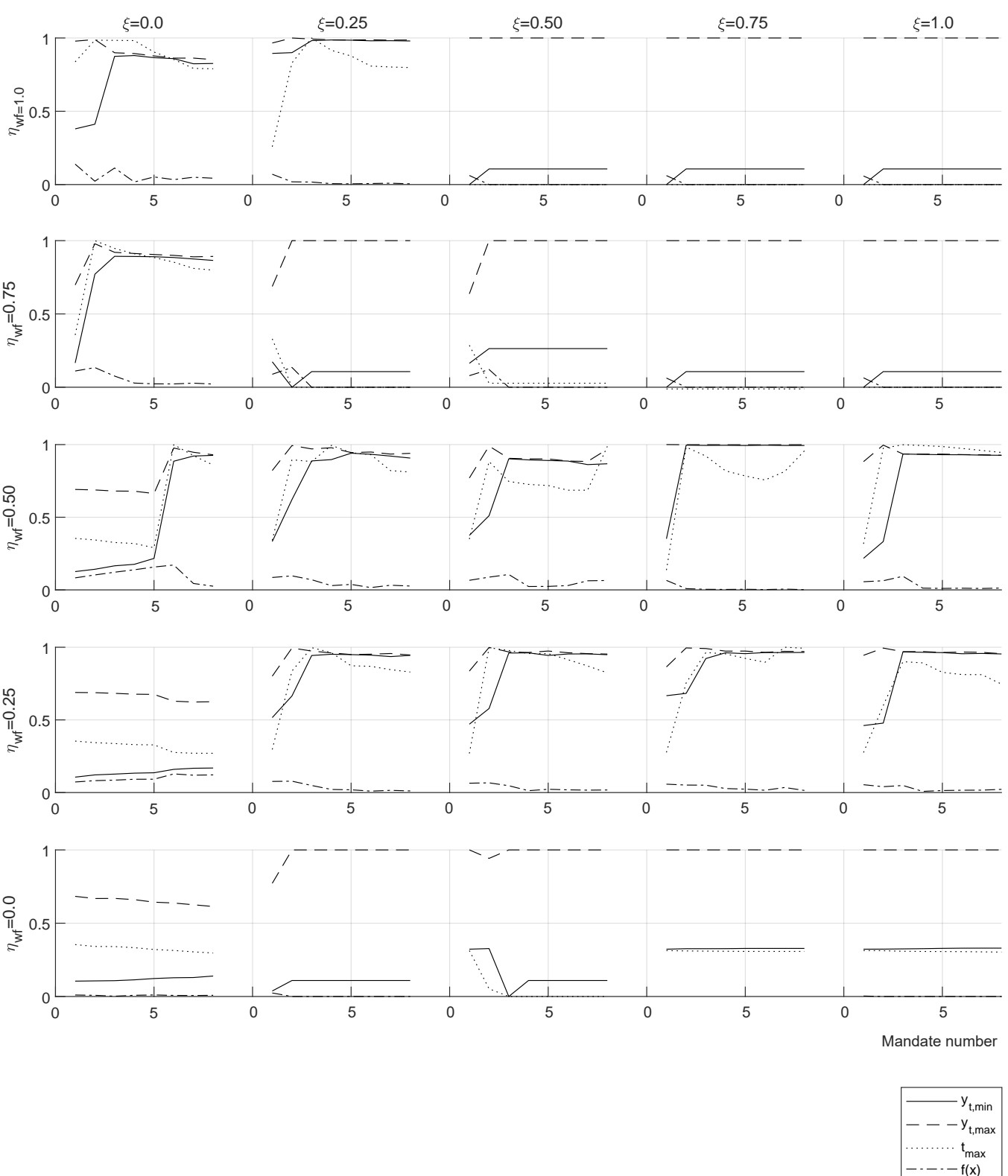

**Figure A5.** Tax policy evolution under yardstick vote during eight mandates for a series of welfare efficiency values $\eta_{wf}$ and tax disincentive $\xi$. Per citizen criterion. Mandate length: 4 years. Voters improvement demand 0%.

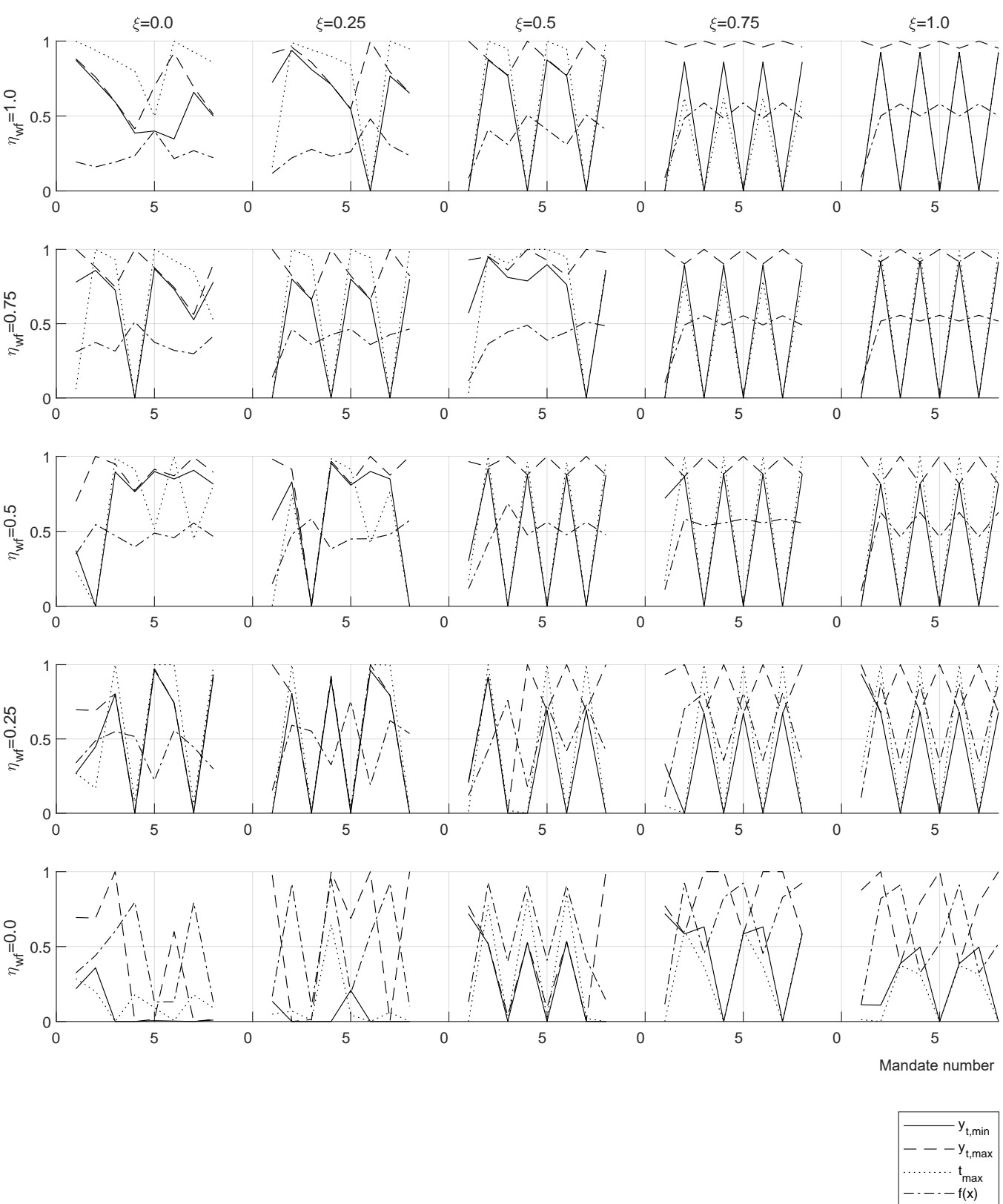

**Figure A6.** Tax policy evolution under yardstick vote during eight mandates for a series of welfare efficiency values $\eta_{wf}$ and tax disincentive $\xi$. Per citizen criterion. Mandate length: 4 years. Voters improvement demand 1%.

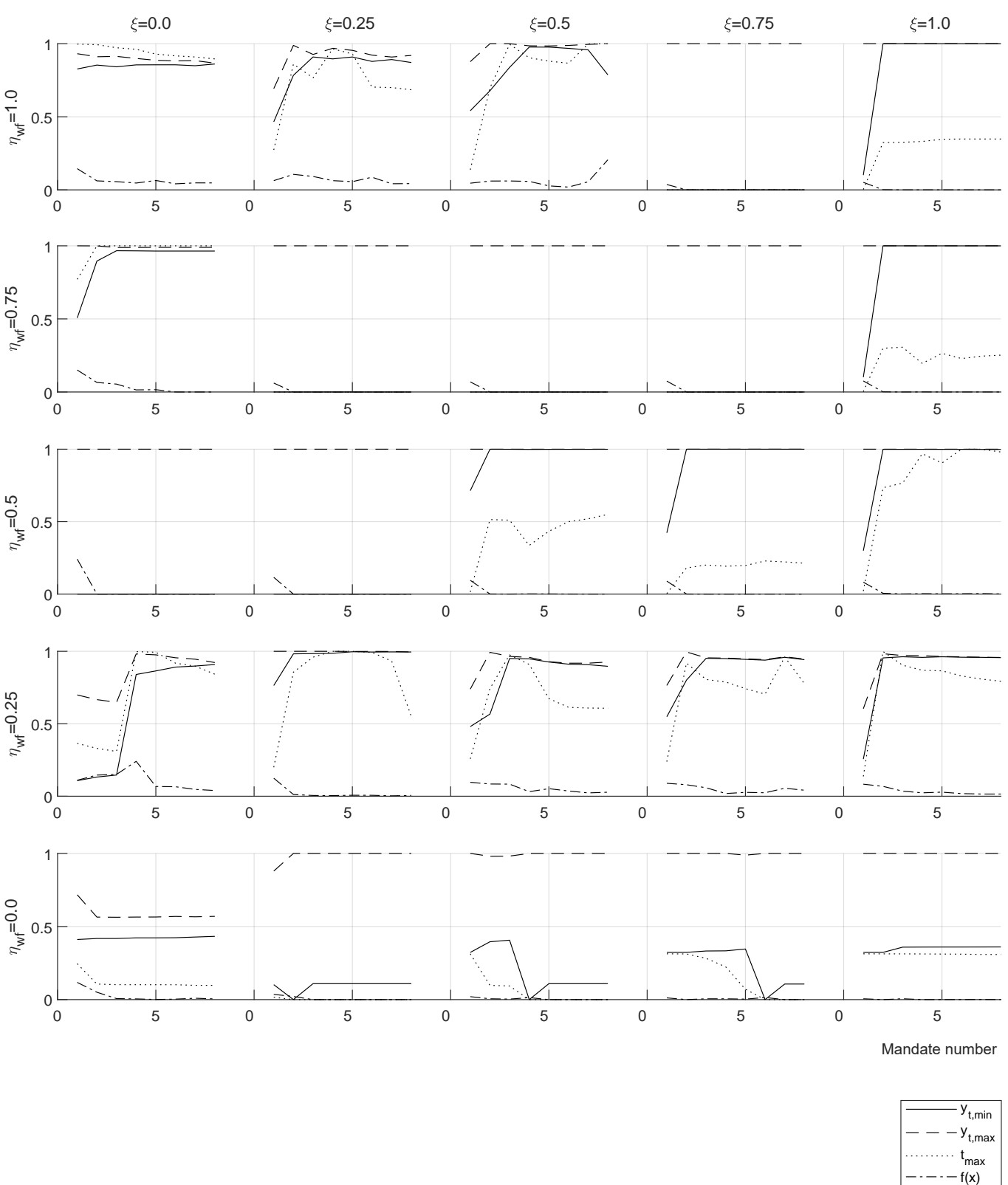

**Figure A7.** Tax policy evolution under yardstick vote during eight mandates for a series of welfare efficiency values $\eta_{wf}$ and tax disincentive $\xi$. Per citizen criterion. Mandate length: 8 years. Voters improvement demand 0%.

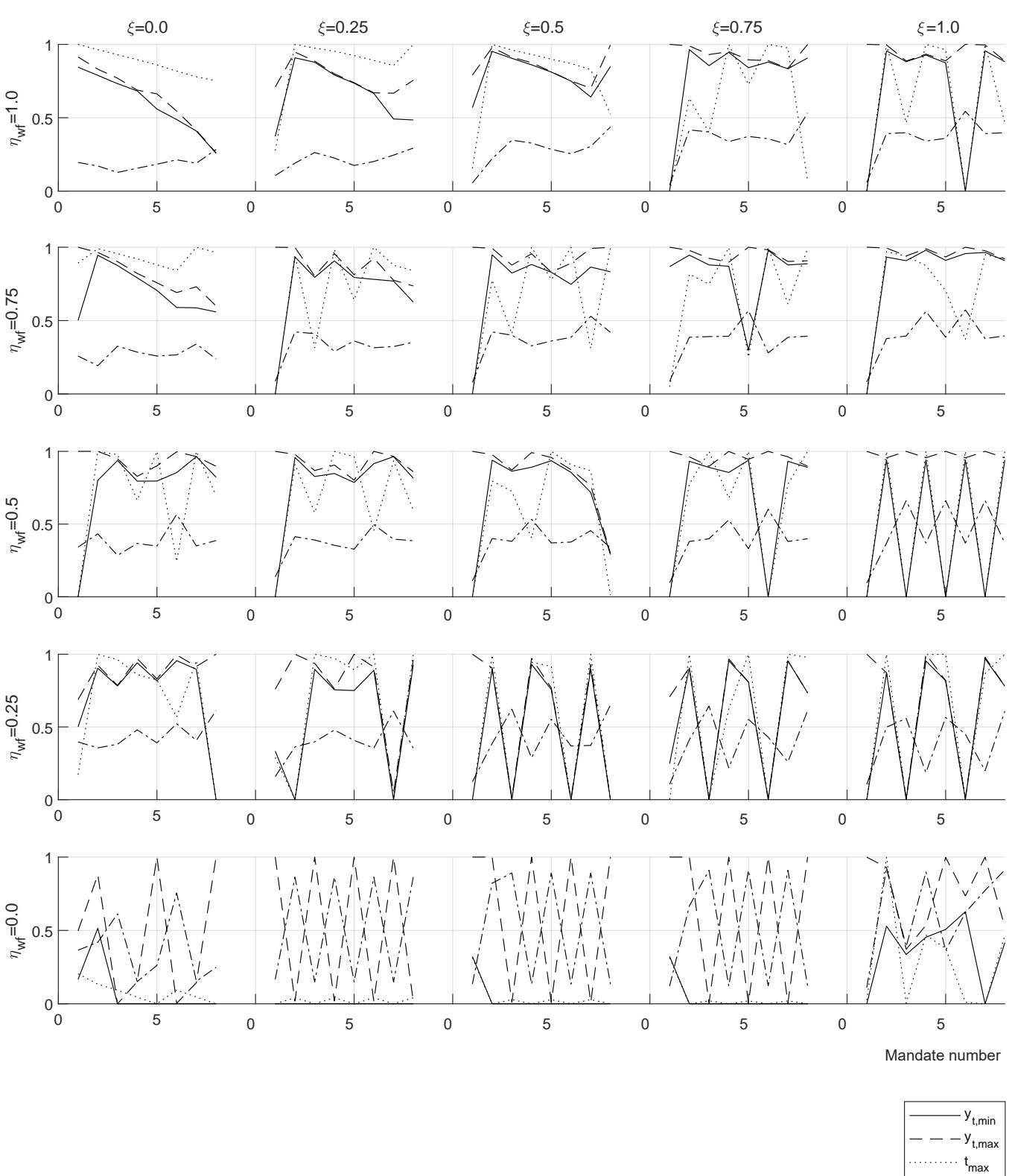

**Figure A8.** Tax policy evolution under yardstick vote during eight mandates for a series of welfare efficiency values $\eta_{wf}$ and tax disincentive $\xi$. Per citizen criterion. Mandate length: 8 years. Voters improvement demand 1%.

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
