# Peer review of "Optimal Policymaking under Yardstick Vote: An Experimental Study"

_games, doi:10.3390/g13030042_

Round 1

Reviewer 1 Report

I am satisfied with this revised manuscript.

All the efforts are worthy!

Reviewer 2 Report

     I have carefully looked at all of the authors' responses to all of my earlier comments and have also checked these responses against the revised manuscript.  The responses are very thorough and do address all of my comments, and the manuscript seems to be greatly improved.  As I indicated in my previous report, though, much of the paper is outside of my areas of specialization, so I am unable to properly assess many facets of the paper.

This manuscript is a resubmission of an earlier submission. The following is a list of the peer review reports and author responses from that submission.

Round 1

Reviewer 1 Report

See attached file.

Reviewer 2 Report

Comment to authrs: 

The authors adopt meta-huristic algorothm to handle the optimization task.

The inherent idea of this paper is uqite interesting and the representation of this this paper is straightforward.

Even this paper poses numerous advantages, I still feel the authors need to add some illustation to strengthen its quality. 

The authors need to explain more why this task can be converted into optimization task, and what is the advantage of handle this task and if we can not handle this task properly, what kind of side effects we confrronted. 

There are so many different meta-huristic algorithms in computer science domain, why the authors adopt PSO?

Please add the advantages of PSO, and demonstrates it effectiveness in experimental outocmes.

Is this a continuous optimization task or discrete optimization task? Please explain.

Please add a managerial section to let readers realize the benefits of adopted this algorithm.

Reviewer 3 Report

I don't think Games is the appropriate outlet for this study.  After pondering it for a few days, I think the authors should consider Journal of Public Economic Theory or Public Finance Review.  There is no real game theory present in the paper, which disqualifies it for publication in Games.  Besides this critical issue, the paper is probably overly lengthy for either of the two journals specified above.
